# SELF-CORRECTING TEXT-TO-VIDEO GENERATION WITH MISALIGNMENT DETECTION AND LOCALIZED REFINEMENT

## ABSTRACT

Recent text-to-video (T2V) diffusion models have made remarkable progress in generating high-quality and diverse videos. However, they often struggle to align with complex text prompts, particularly when multiple objects, attributes, or spatial relations are specified. We introduce **VIDEOREPAIR**, the first self-correcting, training-free, and model-agnostic video refinement framework that automatically detects fine-grained text–video misalignments and performs targeted, localized corrections. Our key insight is that even misaligned videos usually contain correctly rendered regions that should be preserved rather than regenerated. Building on this observation, VIDEOREPAIR proposes a novel region-preserving refinement strategy with three stages: (i) *misalignment detection*, where systematic MLLM-based evaluation with automatically generated spatio-temporal questions identifies faithful and misaligned regions; (ii) *refinement planning*, which preserves correctly generated entities, segments their regions across frames, and constructs targeted prompts for misaligned areas; and (iii) *localized refinement*, which selectively regenerates problematic regions while preserving faithful content through joint optimization of preserved and newly generated areas. This self-correcting, region-preserving strategy converts evaluation signals into actionable guidance for refinement, enabling efficient and interpretable corrections. On two challenging benchmarks, EvalCrafter and T2V-CompBench, VIDEOREPAIR achieves substantial improvements over recent baselines across diverse alignment metrics. Comprehensive ablations further demonstrate the efficiency, robustness, and interpretability of our framework. [1]

## 1 INTRODUCTION

Recent text-to-video (T2V) diffusion models (Ho et al., 2022; Singer et al., 2022; Esser et al., 2023; Blattmann et al., 2023; Khachatryan et al., 2023; Wang et al., 2023a; Yang et al., 2024) have achieved impressive photorealism and versatility across diverse domains. Despite these advances, current models often struggle to faithfully follow input text prompts, especially when the prompt specifies multiple objects and attributes. Typical errors include generating the wrong number of objects, mismatched attribute bindings, or distorted regions.

To mitigate these issues, recent works (Yang & Wang, 2024; Tian et al., 2024) propose compositional T2V techniques that improve text–video alignment. While these methods enhance compositionality, they lack explicit feedback mechanisms to detect and correct misalignments, limiting their adaptability and interpretability in real-world scenarios. In parallel, several image-based studies (Mañas et al., 2024; Wu et al., 2024) have introduced training-free frameworks that refine outputs using guidance from LLMs or MLLMs. However, these approaches are often computationally expensive, dependent on external generators, or prone to visual inconsistencies and such challenges are exacerbated in videos, where temporal coherence is essential.

To address these challenges, we introduce VIDEOREPAIR, the first self-correcting framework for text-to-video generation that is compatible with any diffusion-based T2V backbone and requires

---

[1]Code and qualitative examples are provided in the supplementary materials.

no additional training or architectural changes. Our key insight is that even when generated videos contain misaligned or distorted objects, certain key elements are often accurately generated in specific regions. Similar to how humans revise creative work by fixing only the errors while keeping what is correct, VIDEOREPAIR preserves accurately generated regions and selectively refines only the problematic ones. This region-preserving strategy leverages diffusion models' natural ability to regenerate content from noise while avoiding unnecessary changes to faithful areas. Moreover, detecting correctly generated content via grounding and segmentation is substantially easier than exhaustively enumerating all possible distortions, making our approach both reliable and efficient.

Building on this intuition, VIDEOREPAIR implements region-preserving refinement through three mutually reinforcing processes: *misalignment detection*, *refinement planning*, and *localized refinement* as illustrated in Fig. 1. Unlike prior refinement frameworks that operate on the entire video indiscriminately, VIDEOREPAIR follows a self-correcting, region-preserving paradigm: it distinguishes correctly generated regions from misaligned ones and regenerates only the latter. This transforms evaluation feedback into actionable generative guidance, allowing precise corrections without discarding high-quality content and establishing a new paradigm for efficient, interpretable video refinement. Specifically, spatio-temporal evaluation questions derived from the prompt expose fine-grained errors; these signals guide the selection of entities to preserve and the construction of a targeted refinement prompt; and localized regeneration is then harmonized with preserved regions to yield perceptually seamless videos.

We validate VIDEOREPAIR on two challenging benchmarks, EvalCrafter (Liu et al., 2024b) and T2V-CompBench (Sun et al., 2024a), which cover diverse prompt categories including object counts, spatial relations, and global scene attributes. Empirically, VIDEOREPAIR substantially outperforms existing refinement methods across a wide range of compositional prompts, while preserving global quality aspects such as visual fidelity, motion smoothness, and temporal consistency. We further provide detailed ablations on each component, error accumulation, inference latency, and robustness to different MLLM replacements, underscoring the generality and reliability of our framework.

Our key contributions are as follows:

- We present the first self-correcting, training-free framework for text-to-video generation, compatible with diffusion-based T2V backbones, that detects misalignments via MLLM-based evaluation and plans targeted refinements.

- We introduce a region-preserving refinement strategy that transforms evaluation feedback into actionable guidance, preserving correct regions while selectively regenerating misaligned ones, offering both effective correction and interpretable feedback.

- We show that VIDEOREPAIR consistently improves text–video alignment across diverse models and benchmarks, while maintaining fidelity, temporal coherence, and motion quality, outperforming all prior training-free refinement approaches.

## 2 RELATED WORKS

**Text-to-video generation with diffusion models.** Text-to-video (T2V) diffusion models (Ho et al., 2022; Hong et al., 2022; Singer et al., 2022; Esser et al., 2023; Wu et al., 2023b; Blattmann et al., 2023; Luo et al., 2023) aim to produce videos describing given text prompts. VideoCrafter2 (Chen et al., 2024) synthesizes low-quality videos with high-quality images through a joint training design of spatial and temporal modules, obtaining high-quality videos. T2V-turbo (Li et al., 2024a) presents a distilled video consistency model (Wang et al., 2023c; Song et al., 2023) for improved and rapid video generation. A line of recent work also studies LLM-guided planning frameworks, where an LLM first generates an overall plan (*e.g.*, list of bounding boxes) then video diffusion models render the scene following the plan (Lin et al., 2023; Lian et al., 2024; Lv et al., 2024). However, even the recent T2V diffusion models suffer from misalignment problems. In the following, we discuss the research direction of refining the image/video diffusion models, including VIDEOREPAIR.

**Automatic refinement for image diffusion models.** Recent works propose refinement frameworks that automatically improve diffusion models' text alignment (Sun et al., 2023; Li et al., 2024b; Mañas et al., 2024; Wu et al., 2024). A training-based approaches detect errors of a diffusion model, generates training data, and then finetune the model to improve alignment (Sun et al., 2023; Li et al.,

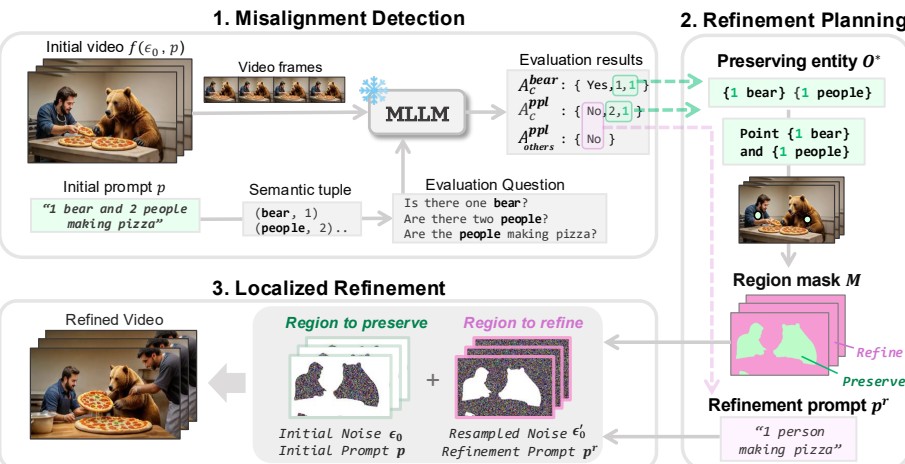

Figure 1: **Illustration of VIDEOREPAIR**. VIDEOREPAIR refines the generated video in three stages: (1) misalignment detection (Sec. 3.1), (2) refinement planning (Sec. 3.2) and (3) localized refinement (Sec. 3.3).

2024b). However, these methods are often expensive and can make the model overfit to specific domains of generated training data. Another line of work proposes training-free framework (Mañas et al., 2024; Wu et al., 2024). OPT2I (Mañas et al., 2024) presents iterative prompt optimization, where an LLM provides various variations of text prompts, T2I diffusion models generate images from the prompts, and the images are ranked with a T2I alignment score (*e.g.*, DSG (Cho et al., 2024)) to provide the final image. Since no explicit feedback is given to the backbone generation model, it usually takes long iterations (*e.g.*, 30 LLM calls) to find a prompt that provides improved alignment, making the framework expensive to use in practice. SLD (Wu et al., 2024) provides more explicit guidance by generating a bounding-box level plan with an LLM, followed by operations such as object addition, deletion, and repositioning. However, SLD depends on an external layout-guided object generator (*e.g.*, GLIGEN (Li et al., 2023)) to insert objects, and the added content often fails to harmonize with the original image. Moreover, while bounding-box operations can be extended to videos, SLD lacks mechanisms to enforce temporal coherence across frames, frequently leading to inconsistent trajectories and motion artifacts. In contrast, VIDEOREPAIR is the first training-free refinement framework that delivers fine-grained localized feedback and is compatible with any T2V diffusion model, without relying on additional generators.

# 3 VIDEOREPAIR

We introduce VIDEOREPAIR, the first *training-free, model-agnostic self-correcting* framework for text-to-video generation. Unlike prior refinement approaches that either use only prompt optimization (Mañas et al., 2024) or rely on external generative models (Wu et al., 2024), our approach follows a new principle: *preserve the correct region, selectively repair where it is wrong*. This principle distinguishes VIDEOREPAIR from generic mask-based inpainting or editing: the preserved regions are determined not by manual masks or heuristic rules, but by an automatic video evaluation and planning process that identifies semantically aligned objects directly from the input prompt and the generated video. By tightly coupling evaluation, planning, and refinement within the same T2V backbone, VIDEOREPAIR enables localized regeneration that improves compositional fidelity while maintaining temporal consistency and visual realism.

**Problem Statement.** Our goal is to improve text-video alignment using a pre-trained T2V diffusion model $f(\cdot)$, without requiring any additional fine-tuning. Given a text prompt $p$ and initial noise $\epsilon_0 \sim \mathcal{N}(0, \boldsymbol{I})$, we generate an initial video $V_0 = f(p, \epsilon_0)$, where $V_0 \in \mathbb{R}^{3 \times H \times W \times T}$ and $H$, $W$, and $T$ denote the height, width, and number of frames, respectively. If $V_0$ exhibits misaligned or inaccurate content, we evaluate it using a set of questions derived from the prompt $p$ and construct a refinement plan. Then, we perform localized self-refinement with the same T2V model $f(\cdot)$, producing a refined video $V_1$ that better aligns with the original prompt. We describe each stage in detail below.

## 3.1 MISALIGNMENT DETECTION

**Generate video evaluation questions.** To diagnose the initial video $V_0$, we generate video evaluation questions from $p$. Unlike prior question-based evaluations in the image domain (Hu et al., 2023; Cho et al., 2024), these questions provide *spatio-temporal feedback signals* that directly guide refinement planning. It goes beyond simple object-existence checks by explicitly capturing *counts, attributes, spatio-temporal relations, actions, and scene-level global properties*, all of which are critical for faithful video-text alignment. Given a prompt $p$, we first extract a semantic tuple $\mathcal{T}$, a structured representation of entities, attributes, relationships, and actions relevant to the video. Using this as guidance, we employ in-context learning with an LLM to generate a set of evaluation questions $Q$. The resulting set $Q$ is divided into two disjoint subsets: $Q_c$ (questions focused on object counting) and $Q_{\text{others}}$ (questions covering all other aspects, such as action, attributes, and scene-level global properties), reflecting the distinct nature of count-based reasoning versus semantic understanding. To better support fine-grained counting, we let the LLM generate count-specific questions for individual objects (e.g., "Is there *one* bear?") rather than merely verifying object existence (e.g., "Is there a bear?"). Our ablation study (see Tab. 3) demonstrates that these evaluation questions provide more effective refinement guidance compared to existing question-based evaluation methods. Additional implementation details are provided in the Appendix.

**Answering to identify video errors.** We now evaluate $V_0$ to determine which region requires refinement, as illustrated in Fig. 1 (top right). Given the entity set $O$ (*i.e.*, object or scene element) from $\mathcal{T}$, we group $Q^o = \{Q_c^o, Q_{\text{others}}^o\} \subset Q$ as the subset of questions that contain the name of $O$ Note that this entity captures not only localized object discrepancies but also global misalignments between $p$ and $V_0$ To this end, we employ an MLLM to answer each predefined question set $Q^o$ with binary judgments. For count-related questions $Q_c^o$, we prompt the model to output both a binary decision and an estimated object count, resulting in a triplet $A_c^o = \{b_c^o, n_p^o, n_v^o\}$, where $n_p^o$ and $n_v^o$ denote the number of instances of object $o$ in the prompt $p$ and the video $V_0$, respectively. The binary answer $b_c^o$ is set to 1 if $n_p^o = n_v^o$, and 0 otherwise. For example, in Fig. 1, the question "Is there one bear?" results in $b_c^{\text{bear}} = 1$ when both the prompt and the video indicate a single bear (i.e., $n_p^{\text{bear}} = n_v^{\text{bear}} = 1$). For other type of questions $Q_{\text{others}}^o$, we prompt the model to return only a binary response $A_{\text{others}}^o = \{b_{\text{others}}^o\}$, where $b_{\text{others}}^o = 1$ indicates alignment between the element in $V_0$ and $p$, and $b_{\text{others}}^o = 0$ otherwise. If an entity disappears or becomes distorted across frames, we also regard it as a misalignment case. We aggregate binary evaluation results into a video-level accuracy score in the range $[0, 1]$. If the score is $1.0$, we terminate the process early, as the initial video is already fully correct. If the score is $0.0$, we instead re-generate the video with a new random seed to avoid uninformative outputs.

## 3.2 REFINEMENT PLANNING

**Identifying visual content to retain.** As mentioned earlier, VIDEOREPAIR aims to retain accurately generated regions in the initial video while correcting only the mis-generated ones to ensure improved text-video alignment. To this end, we first identify the key entity $O^*$ and determine the number $N^*$ of its instances to be preserved. To select which entity should be retained, we prompt the MLLM with question–answer pairs and $V_0$ as input, allowing it to identify correctly generated entities to preserve. For countable entities, the number of preserved instances $N^*$ is determined from the triplet $A_c^{o^*} = \{b_c^{o^*}, n_p^{o^*}, n_v^{o^*}\}$ as

$$N^* = \begin{cases} n_p^{o^*} & \text{if } n_p^{o^*} \le n_v^{o^*}, \\ n_v^{o^*} & \text{otherwise,} \end{cases} \tag{1}$$

where $n_p^{o^*} < n_v^{o^*}$ indicates that excess instances should be removed, and $n_p^{o^*} > n_v^{o^*}$ suggests that additional instances are required. For example, in Fig. 1, if $O^*$ represents people with $n_p^{o^*} = 2$ and $n_v^{o^*} = 1$, we set $N^* = 1$ to preserve one person. Note that multiple instances of $O^*$ may exist if there are several plausible entities to retain. For global scene elements (e.g., background), which are not inherently countable, we treat $N^*$ as a presence indicator, setting $N^* = 1$ if the element is preserved. This unified notation allows us to consistently handle both entity- and scene-level preservation within the same refinement planning framework.

**Identifying regions to preserve.** Based on the entity selection $O^*$, we localize the regions corresponding to correctly generated content within the video frames, as shown in Fig. 1 (top right). For countable entities, given the set of preserved instances $O^*$ and their quantities $N^*$, we first construct a pointing prompt using the template: "Point the biggest $\{N^*\}$ $\{O^*\}$" (e.g., "Point the biggest 1 bear"). This prompt is used to obtain 2D coordinates indicating the spatial locations of $O^*$ in each sampled frame. Using these coordinates as initialization, we apply a segmentation model to extract entity-specific regions, resulting in binary segmentation masks $\mathbf{M} \in \mathbb{R}^{H \times W \times T}$ that preserve the correctly generated entities. In practice, for global elements such as background, we simply preserve the entire frame region or assign a broad background mask if the property is rendered correctly. By combining these with the region-level masks, we obtain a dense, frame-aligned segmentation map $\mathbf{M}$ that preserves both entity- and scene-level regions.

**Prompt regeneration for regions requiring refinement.** We additionally generate a local prompt for refinement to enable distinct control over different regions during generation. To this end, we prompt an LLM to produce a refinement-oriented prompt, $p^r$, based on $Q$ but excluding any questions related to $O^*$. As illustrated in Fig. 1 step 1, this regenerated local prompt will be used to guide the denoising process for specific areas to be refined during video generation in a later stage (will be discussed in Sec. 3.3).

### 3.3 LOCALIZED REFINEMENT

At this stage, we refine the video to improve alignment while preserving coherence with the original content. While video editing (Jiang et al., 2025; Yang et al., 2025) preserves masked regions and enforces visual consistency, it is limited in its ability to freely introduce or correct entities misaligned with the original prompts. Similarly, inpainting (Lugmayr et al., 2022; Bian et al., 2025) fills missing regions with locally consistent textures but lacks mechanisms for semantically guided object introduction or correction from textual input. (See Tab. 3) Instead of these approaches, we selectively re-initialize noise only in misaligned regions and apply distinct text prompts to preserved and refined areas, enabling targeted corrections while maintaining overall video consistency.

**Localized noise re-initialization.** We adopt a mask-based strategy in which only regions marked for refinement are re-initialized with newly sampled noise $\epsilon_0' \sim \mathcal{N}(0, \mathbf{I})$, while preserved regions retain their original noise $\epsilon_0$. This selective resampling maintains consistency in faithful areas while allowing controlled updates in misaligned ones. To transform the pixel-level, multi-frame mask $\mathbf{M}$ into the latent space, we apply block averaging (pooling), yielding a hybrid noise map:

$$\epsilon_0^* = (\epsilon_0 \otimes \text{pool}(\mathbf{M}, d)) + (\epsilon_0' \otimes (1 - \text{pool}(\mathbf{M}, d))), \tag{2}$$

where $\text{pool}(\cdot, d)$ downsamples the mask and $\otimes$ denotes element-wise multiplication. This noise map $\epsilon_0^*$ is then used with localized prompts to guide the frozen diffusion model.

**Localized text guidance.** Afterward, we apply distinct text prompts to different spatial regions of the video based on their noise re-initialization status, using the binary segmentation mask $\mathbf{M}$ to separate preserved ($M_{\text{pres}}$) and re-initialized ($M_{\text{refine}} = 1 - M_{\text{pres}}$) areas. For the re-initialized regions, we guide generation in the latent space using regenerated prompts $p^r$ (See Sec. 3.2) tailored to those areas. In parallel, motivated by recent findings on noise bias (Sun et al., 2024b; Ban et al., 2024; Qi et al., 2024), we reuse the original prompt $p$ to preserve features related to $O^*$ in the retained regions. This regionalized decomposition of the original prompt allows for the addition or modification of objects in re-initialized areas, while maintaining the integrity of correctly generated content in preserved regions.

**Harmony with original elements.** To further ensure global coherence between preserved and refined regions, we regenerate all pixels through two separate diffusion paths and fuse them via joint optimization. Specifically, at each denoising step $t$, we run the diffusion model $f(\cdot)$ twice with different prompts and noises: $\hat{V}_{\text{pres}} = f(V_t, p, \epsilon_0)$ for the preserved regions, and $\hat{V}_{\text{refine}} = f(V_t, p^r, \epsilon_0')$ for the refined regions. The final fused output $\tilde{V}$ is obtained by solving:

$$V_1 = \arg\min_{\tilde{V}} \left\| M_{\text{pres}} \otimes (\tilde{V} - \hat{V}_{\text{pres}}) \right\|^2 + \left\| M_{\text{refine}} \otimes (\tilde{V} - \hat{V}_{\text{refine}}) \right\|^2, \tag{3}$$

Table 1: **Evaluation results on EvalCrafter with other baselines.** Note that we focus on these four splits, whereas the official website reports the average across all splits. We highlight the quality and consistency performance in red if it deteriorates by more than 1% from the original performance.

| Method | Text-Video Alignment | | | | | Visual Quality | Motion Quality | Temporal Consistency |
|---|---|---|---|---|---|---|---|---|
| | Count | Color | Action | Others | Average | | | |
| VideoCrafter2 | 47.52 | 46.28 | 44.07 | 46.02 | 45.97 | 61.8 | 62.6 | 62.9 |
| + LLM paraphrasing | 45.87 | 47.81 | 44.41 | 45.16 | 45.81 | 62.4 | 62.7 | 62.7 |
| + SLD (Wu et al., 2024) | 44.47 | 46.45 | 39.89 | 44.06 | 43.72 | 52.5 | 62.2 | 44.4 |
| + OPT2I (Mañas et al., 2024) | 47.69 | 47.67 | 45.04 | 44.65 | 46.26 | 62.1 | 62.6 | 62.8 |
| + VIDEOREPAIR (Ours) | **49.84** | **51.57** | **45.78** | **48.12** | **48.83** | 62.1 | 62.4 | 62.0 |
| T2V-turbo | 46.14 | 43.06 | 41.42 | 43.16 | 43.94 | 63.3 | 57.8 | 61.6 |
| + LLM paraphrasing | 49.49 | 43.16 | 41.32 | 44.75 | 44.68 | 62.9 | 52.9 | 61.9 |
| + SLD (Wu et al., 2024) | 47.39 | 43.99 | 42.13 | 43.28 | 44.20 | 56.6 | 58.2 | 49.2 |
| + OPT2I (Mañas et al., 2024) | 47.44 | 45.00 | 44.64 | **45.54** | 45.66 | 63.3 | 56.4 | 48.9 |
| + VIDEOREPAIR (Ours) | **51.27** | **46.66** | **45.81** | 45.45 | **47.30** | 63.2 | 57.9 | 61.8 |
| CogVideoX-5B | 47.88 | 49.63 | 37.76 | 44.78 | 45.01 | 65.8 | 61.0 | 61.8 |
| + LLM paraphrasing | 45.58 | 46.56 | 37.17 | 43.18 | 43.12 | 58.4 | 61.1 | 61.7 |
| + SLD (Wu et al., 2024) | 47.73 | 46.27 | 39.55 | 43.75 | 44.33 | 49.6 | 51.2 | 21.0 |
| + OPT2I (Mañas et al., 2024) | 48.62 | 48.89 | **41.39** | 43.62 | 45.63 | 59.7 | 60.9 | 61.9 |
| + VIDEOREPAIR (Ours) | **49.63** | **49.94** | 40.69 | **45.36** | **46.41** | 64.8 | 61.1 | 61.9 |

This joint optimization allows $\tilde{V}$ to seamlessly blend preserved and refined regions, reducing mismatches at region boundaries and producing perceptually smooth, globally coherent videos.

**Video ranking.** Similar to generating multiple candidate prompts in (Mañas et al., 2024), we produce $K$ refined videos using different random seeds and select the best one based on our video scores, as obtained in Sec. 3.1, thus avoiding additional computations or resource burdens. If multiple videos receive a tied video score, we select the video with the highest BLIP-BLEU score (Liu et al., 2024b) among them. Note that VIDEOREPAIR does not depend on ranking process (see Tab. 4)

## 4 EXPERIMENTS

### 4.1 EXPERIMENT SETUPS

**Benchmarks and evaluation metrics.** We evaluate our method on two text-to-video generation benchmarks: EvalCrafter (Liu et al., 2024b) and T2V-CompBench (Sun et al., 2024a). [2]

**(1) EvalCrafter.** We follow the official metadata[3] to split prompts by attributes into four sections: *count*, *color*, *action*, and *others*. The *others* category includes scenery-related prompts, such as Camera movement (e.g., *"Zoom in"*), Landscape (e.g., *"A bustling street in Paris"*), and Style (e.g., *"Polaroid style"*). For evaluation metrics, we report four groups: text-video alignment, video quality, motion quality, and temporal consistency. Details of the evaluation metrics are provided in the appendix.

**(2) T2V-CompBench (ver.1).** We adopt three compositional reasoning categories from this benchmark: spatial relationships, generative numeracy, and consistent attribute binding, with 100 prompts in each category. We use ImageGrid-LLaVA (Liu et al., 2024a) for consistent attribute binding evaluation and GroundingDINO (Liu et al., 2023) for the other two dimensions. Further details are provided in the appendix.

**Implementation details.** We implement VIDEOREPAIR on three recent T2V models: T2V-turbo (Li et al., 2024a), VideoCrafter2 (Chen et al., 2024), and CogVideoX-5B (Yang et al., 2024). T2V-Turbo and VideoCrafter2 generate 16 frames, while CogVideoX-5B generates 81 frames. Unless otherwise specified, all experiments use $K = 5$ with a single iteration. For MLLM and LLM components, we primarily use GPT-4o, and for segmentation we employ Semantic-SAM (L). Additional details are provided in the appendix.

---

[2]All reported results are based on our own experiments.

[3]https://github.com/evalcrafter/EvalCrafter/blob/master/metadata.json

**Baselines.** We compare VIDEOREPAIR against recent refinement frameworks, OPT2I (Mañas et al., 2024) and SLD (Wu et al., 2024), on the same three T2V models described above. Although these baselines were originally proposed for text-to-image refinement, we extend their implementations to the video setting. For OPT2I, we score the videos using the original DSG (Cho et al., 2024) and iteratively generate five prompt candidates. For SLD, since its refinement model is based on LMD+ (Lian et al., 2023), we apply SLD frame-by-frame to the initial outputs of T2V models. We also include *LLM paraphrasing* as a baseline, where GPT-4 generates diverse paraphrases of the initial prompt. To ensure fairness, we unify random seeds across all experiments so that all methods refine the same initial videos. In addition, we compare VIDEOREPAIR with state-of-the-art (SoTA) T2V models, including ModelScope (Wang et al., 2023a), ZeroScope (huggingface, 2023), Latte (Ma et al., 2024), Show-1 (Zhang et al., 2024), Open-Sora-Plan v1.1.0 (PKU-Yuan Lab etc, 2023), VideoTetris (Tian et al., 2024), and Vico (Yang & Wang, 2024). Further details are provided in the appendix.

## 4.2 QUANTITATIVE RESULTS

As shown in Tab. 1 and Tab. 2, VIDEOREPAIR consistently outperforms other refinement baselines (OPT2I, SLD) as well as strong compositional T2V models (e.g., Vico, VideoTetris) across both benchmarks. On EvalCrafter, VIDEOREPAIR achieves relative alignment gains of **+6.22%**, **+7.65%**, and **+3.11%** over VideoCrafter2, T2V-turbo, and CogVideoX-5B, respectively. For the *Other* section, we further break down the results into camera movement, landscape, and style prompts; see Tab. 6 in the appendix for details. On T2V-CompBench, VIDEOREPAIR fur-

Table 2: **Evaluation results on T2V-CompBench.**

| Method | Consist-Attr | Spatial | Numeracy | Avg. |
|---|---|---|---|---|
| ModelScope (Wang et al., 2023a) | 0.5148 | 0.4118 | 0.1986 | 0.3750 |
| ZeroScope (huggingface, 2023) | 0.4011 | 0.4287 | 0.2408 | 0.3568 |
| Latte (Ma et al., 2024) | 0.4713 | 0.4340 | 0.2320 | 0.3791 |
| Show-1 (Zhang et al., 2024) | 0.5670 | 0.4544 | 0.3086 | 0.4433 |
| Open-Sora-Plan (PKU-Yuan Lab etc, 2023) | 0.4246 | 0.4520 | 0.2331 | 0.3699 |
| Vico (Yang & Wang, 2024) | 0.6470 | 0.5425 | 0.2762 | 0.4886 |
| VideoTetris (Tian et al., 2024) | 0.6211 | 0.4832 | 0.3467 | 0.4836 |
| VideoCrafter2 | 0.6812 | 0.5214 | 0.2906 | 0.4977 |
| + VIDEOREPAIR (Ours) | **0.7275** | **0.5690** | **0.3278** | **0.5383** |
| T2V-turbo | 0.7025 | 0.5492 | 0.2496 | 0.5004 |
| + VIDEOREPAIR (Ours) | **0.7675** | **0.5807** | **0.2709** | **0.5439** |
| CogVideoX-5B | 0.6220 | 0.4988 | 0.2228 | 0.4479 |
| + VIDEOREPAIR (Ours) | **0.6725** | **0.5811** | **0.3034** | **0.5190** |

ther improves spatial relationships, numeracy, and attribute binding, with relative gains of **+8.16%**, **+8.69%**, and **+15.87%**. These results demonstrate that VIDEOREPAIR effectively corrects fine-grained spatial and temporal misalignments while preserving visual fidelity (std. deviation of visual quality scores: 0.55), establishing a robust refinement framework across diverse T2V models and benchmarks. By contrast, SLD underperforms particularly in the *action* and *count* categories because its frame-level latent fusion fails to maintain consistent object counts and spatial layouts over time. OPT2I shows only modest gains: although it searches optimized prompts with DSG, it operates entirely in text space without spatial or temporal guidance, limiting its ability to fix localized misalignments. Unlike these approaches, VIDEOREPAIR directly evaluates and refines misaligned regions, enabling targeted corrections without degrading overall video quality.

## 4.3 QUALITATIVE RESULTS

Fig. 3 presents qualitative comparisons of other refinement frameworks (OPT2I, SLD, and VIDEOREPAIR) applied to three T2V models. These examples demonstrate the effectiveness of VIDEOREPAIR in addressing *object and attribute misalignment* more reliably than existing methods. In the leftmost example from VideoCrafter2, VIDEOREPAIR accurately generates the specified color attribute (*green* horse) while preserving the running dog. In the middle example from T2V-turbo, VIDEOREPAIR improves the incorrect count of *three dogs*, whereas other baselines either fail to do so (OPT2I) or introduce artificial distortions (SLD). Finally, in the rightmost example from CogVideoX-5B, VIDEOREPAIR successfully captures spatial relationships (positioning the car *behind* the pig) while maintaining the integrity of multi-object generation.

## 4.4 ADDITIONAL ANALYSIS

In this section, we conduct ablations on model components and ranking, analyze error accumulation with human oracle and Qwen2.5-VL (7B) backbones, and present iterative refinement results to

Table 3: **Ablations of different VIDEOREPAIR components.**

| Eval. Question (Sec. 3.1) | Planning (Sec. 3.2) | Video Refinement (Sec. 3.3) | Avg. |
|---|---|---|---|
| - | - | - | 43.54 |
| DSG | random | Ours | 45.18 |
| Ours | random | Ours | 46.92 |
| Ours | $M, p^r$ | Ours | **47.91** |
| Ours | $M$ | VideoGrain | 40.52 |
| Ours | $M$ | VACE | 46.77 |
| Ours | $M, p^r$ | VACE | 44.88 |

Table 4: **Ablations on the video ranking.** We vary the number of video candidates (K) and metrics during the video ranking.

| Method | # of video candidates (K) | Ranking metric | Text-Video Alignment |
|---|---|---|---|
| T2V-turbo | - | - | 43.54 |
| | 5 | Ours | 44.68 |
| + VIDEOREPAIR | 1 | Ours | 46.53 |
| | 5 | Ours | **47.91** |
| | 5 | CLIP | 45.85 |
| | 5 | BLIP-BLEU | 46.77 |

Table 5: **Error accumulation analysis** comparing GPT4o with a human-annotated oracle and Qwen2.5-VL (7B).

| Video Eval. (Sec. 3.1) | Planning (Sec. 3.2) | Count | Color | Action | Avg. |
|---|---|---|---|---|---|
| Human | Human | **52.54** | **49.91** | 44.69 | **49.05** |
| Human | GPT4o | 51.79 | 48.94 | 44.66 | 48.46 |
| GPT4o | GPT4o | 51.27 | 46.66 | 45.81 | 47.91 |
| Qwen | GPT4o | 50.23 | 48.08 | **47.53** | 48.61 |
| Qwen | Qwen | 47.84 | 46.64 | 44.37 | 46.28 |

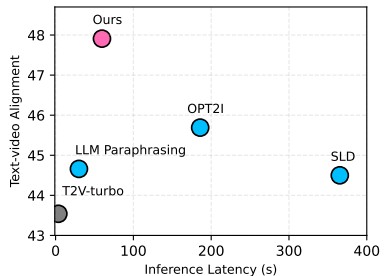

Figure 2: **Inference latency** compared with refinement baselines.

mitigate propagated errors. All ablations are conducted on the *Count*, *Color*, and *Action* splits of EvalCrafter using T2V-turbo, and we report the average text-video alignment performance across these three categories. In each table, our default setup is highlighted with a **blue background**.

**Ablations on the component of VIDEOREPAIR.** In Tab. 3, we analyze the contributions of different components of VIDEOREPAIR, including the type of evaluation questions, planning strategy, and refinement method. For evaluation questions (Sec. 3.1), we compare the original DSG questions with ours. For object selection (Sec. 3.2), we evaluate random planning against our approach that leverages $M$ and $p^r$. For video refinement (Sec. 3.3), we assess the effectiveness of our localized refinement module by comparing it with state-of-the-art Video-to-Video (V2V) editing models (Jiang et al., 2025; Yang et al., 2025), using our planning outputs ($M$ and $p^r$) as editing guidance. Specifically, using $M$ alone corresponds to masked V2V editing, while using both $M$ and $p^r$ incorporates GPT4o to generate an editing prompt from $p^r$, which is then provided to the V2V masked editing model. Overall, the results confirm that the combination of our evaluation questions, planning with $M$ and $p^r$, and localized refinement achieves the best performance, surpassing both random planning and existing V2V editing baselines.

**Ablations on the video ranking.** Tab. 4 shows the effect of varying the number of video candidates (K) and the choice of ranking metric. While simply increasing $K$ from the initial T2V-turbo output does not yield consistent gains (44.68 vs. 43.54). Using alternative metrics such as CLIP or BLIP-BLEU also improves over the baseline but falls short of VIDEOREPAIR, highlighting that VIDEOREPAIR provides more reliable guidance for ranking candidate videos.

**Error accumulation.** We analyze the error accumulation replacing VIDEOREPAIR's components with varying evaluation and planning parts with human annotation and Qwen2.5-VL (7B) (Bai et al., 2025). As shown in Tab. 5, replacing the oracle with GPT4o yields only a modest drop (49.05 → 48.46, 47.91). Interestingly, combining Qwen2.5-VL for video evaluation with GPT4o for planning achieves competitive results (48.61), even outperforming the configuration where GPT4o is used for both steps in the action split. These results highlight the robustness and modularity of our framework, showing that it can reliably operate even with weaker open-source models while flexibly benefiting from stronger ones when available.

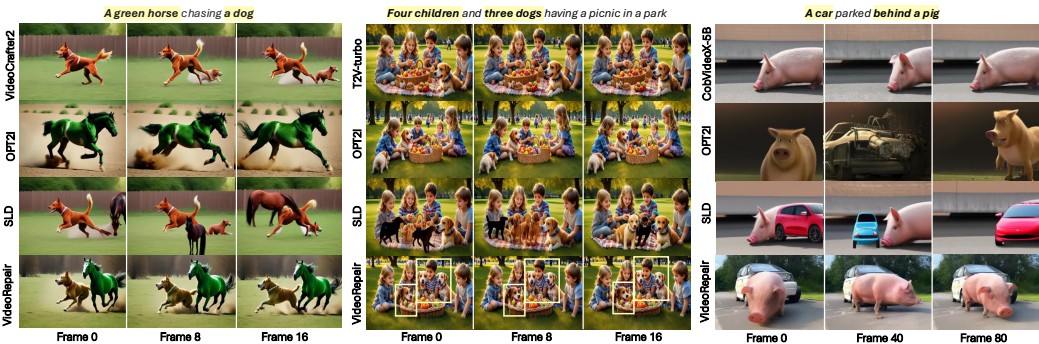

Figure 3: **Videos generated from T2V-turbo, VideoCrafter2 and CogVideoX with refinement frameworks (OPT2I / SLD / VIDEOREPAIR).** VIDEOREPAIR successfully addresses *object and attribute misalignment* issues (e.g., numeracy, spatial relationship, attribute blending) compared to T2V-turbo and other refinement methods.

**Computational cost.** Fig. 2 compares text-video alignment scores against inference latency for different refinement frameworks. OPT2I and SLD incur 3–5× longer runtimes without surpassing our performance. In contrast, VIDEOREPAIR offers the best trade-off between accuracy and efficiency with moderate computational overhead.

**Iterative refinement.** We further explore iterative refinement to progressively enhance text-video alignment, as a single refinement step of VIDEOREPAIR may not fully resolve all inconsistencies with the prompt. As illustrated in Fig. 4, the first refinement partially corrects the misalignment by generating a scene of *a night of camping under the stars*, but some family members disappear. In the second iteration, VIDEOREPAIR recovers all four family members while preserving the rest of the scene. Similarly, in the bottom example of Fig. 4, iterative refinement successfully produces the intended output of seven puppies. Additional qualitative examples are provided in the appendix.

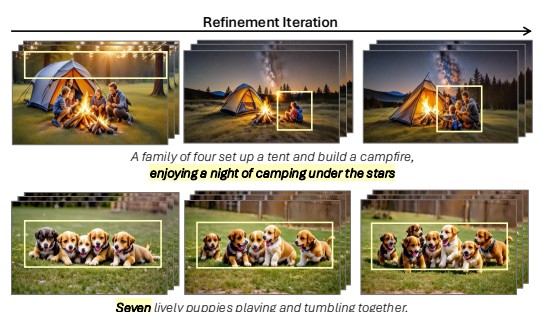

Figure 4: **The iterative refinement with VIDEOREPAIR.** Videos in each column represent the outputs of successive refinement iterations, where the output from the previous step serves as the input.

## 5 CONCLUSION

We propose VIDEOREPAIR, a novel training-free, model-agnostic video refinement framework that improves T2V alignment through automatic detection and correction of fine-grained misalignments. VIDEOREPAIR operates in three stages: (1) misalignment detection identifies misalignments by generating evaluation questions and answers. Next, (2) refinement planning identify key objects to preserve and produces localized prompts for refinement. Finally, (3) localized refinement step enables targeted regeneration of misaligned areas while maintaining accurate content. VIDEOREPAIR substantially outperforms recent baselines across various text-video alignment metrics. We provide a comprehensive analysis of VIDEOREPAIR components and qualitative examples.

### REPRODUCIBILITY STATEMENT

We provide the data and code in the supplementary. Details of the data and model implementations, as well as all hyperparameters, can be found in the Appendix.

ETHICS STATEMENTS

We do not foresee any ethical implications beyond standard ethical and safety considerations that apply to AI research generally.

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

## APPENDIX

## A VIDEOREPAIR IMPLEMENTATION DETAILS

### A.1 DETAILS OF VIDEO EVALUATION QUESTION GENERATION

Given the initial prompt, we construct a semantic tuple $\mathcal{T}$, similar to DSG (Cho et al., 2024) that contains *entities (subjects)*, *attributes*, and *relationships*. Here, attributes are expressed in 2-tuples (subjects, its attribute, (*e.g.*, {bed, blue}), and relationships are in 3-tuples (subject entity, object entity, and their relationship, (*e.g.*, {people, pizza, make}). Based on $\mathcal{T}$, which covers all scene-relevant information, we generate questions $Q$ using *GPT-4-0125* (OpenAI, 2024). Note that although DSG does not account for object counts by design, we can incorporate assessments for whether the generated videos contain the correct number of target objects, thereby guiding automatic refinement with greater accuracy. For example, given a prompt 'there is a bear', DSG only generates an evaluation question "is there a bear?", which only checks the bear's existence, but does not penalize when more than one bear is generated.

### A.2 VIDEO OBJECT EVALUATION

To evaluate the generated videos, we utilize GPT-4o to answer both count-related ($Q_c^o$) and attribute-related ($Q_a^o$) questions, as illustrated in Fig. 23. For $Q_c^o$ prompts, we guide GPT-4o through four steps: reasoning, answering, counting the predicted number of objects ($n_p^o$), and verifying the true count

Figure 5: **Comparison of different refinement methods for alignment.** (a) Prompt optimization (Mañas et al., 2024) presents LLM-based prompt rewriting without visual/fine-grained feedback. (b) Localized feedback method (Wu et al., 2024) supports visual guidance but relies on an external layout-guided generation module. (c) **VIDEOREPAIR** is a training-free, model-agnostic refinement method with localized visual guidance based on automatic video evaluation.

$(n_v^o)$. These steps yield an answer triplet $A_c^o = \{b_c^o, n_p^o, n_v^o\}$. To ensure valid responses, we account for dependencies among questions, following the methodology of DSG (Cho et al., 2024). Each question is sequentially presented to GPT-4o, and the video score is computed as the proportion of correctly answered binary questions across all VQA tasks. If the video score reaches 1.0 (indicating a perfect score), the VIDEOREPAIR process is terminated.

## A.3 KEY OBJECT EXTRACTION

To extract the key concept $O^*$ from the initial videos $V_0$, we sampled frames of $V_0$ and the list of question-answer pairs for each object to GPT4o as shown in Fig. 24. Here, we prioritize selecting objects with a higher number of 1.0 video scores. Moreover, we force GPT4o to select 'object' instead of 'background' elements to improve the accuracy of region decomposition by pointing.

## A.4 REFINEMENT PROMPT GENERATION

To produce a refinement prompt $p^r$, we use GPT4 with instruction as shown in Fig. 25. After getting $O^*$, we can decompose the whole question set $Q$ as $Q^{o*}$ and others depending on whether the $O^*$ keyword is included in the question. To generate $p^r$ from specific question sets, we utilize five manually crafted in-context examples to ensure the accuracy of the generation process. If the video score is 0.0 (indicating a complete failure from VQA) and the key object $O^*$ cannot be identified, we consider the T2V model to have failed in generating any object correctly. In such cases, we paraphrase $Q$ directly into $p^r$ using a large language model (LLM).

## B ADDITIONAL BASELINE DETAILS

**LLM Paraphrasing.** Following (Mañas et al., 2024), we compare VIDEOREPAIR with paraphrasing prompts from LLM. Here, we ask GPT4 to generate diverse paraphrases of each prompt, without any context about the consistency of the images generated from it. The prompt used to obtain paraphrases is provided in Fig. 26.

**OPT2I.** Since OPT2I (Mañas et al., 2024) aims to improve text-image consistency for T2I models, we reimplement OPT2I for T2V setup. Specifically, we replace the original T2I model part with T2V models (T2V-Turbo and VideoCrafter2) to generate outputs. Using GPT-4o, we then pose DSG questions to these outputs. For prompts, we directly adopt the ones provided in the original OPT2I paper. For LLM, we use GPT4 as VIDEOREPAIR. Finally, we perform iterative refinement, running 10 iterations for T2V-Turbo and 5 iterations for VideoCrafter2, with five video candidates per iteration.

**SLD.** To adapt SLD (Wu et al., 2024) to the T2V setup, we apply their official code to individual video frames and maintain their default setup. Note that SLD is a GLIGEN (Li et al., 2023)-based T2I model, which poses challenges for direct extension to video generation. Since SLD operates using DDIM inversion, we use the initial videos generated by T2V-Turbo and VideoCrafter2 as inputs,

enabling the implementation of their noise composition method. Here, we use one iteration for SLD and GPT4 for LLM.

## C ADDITIONAL EVALUATION DETAILS

**EvalCrafter.** To evaluate the effectiveness of VIDEOREPAIR across different prompt dimensions, we decompose EvalCrafter (Liu et al., 2024b) using the official `metadata.json`. Specifically, we utilize the `attributes` key for each prompt and categorize the dataset into 'count', 'color', 'action', 'text', 'face', and 'amp (camera motion)'. Prompts without explicit attributes are grouped into an 'others' category. Among these dimensions, we focus on 'count', 'color', 'action', and 'others', excluding 'text', 'face', and 'amp'. This decision is based on our observation that video errors related to text prompts (*e.g.*, *"the words 'KEEP OFF THE GRASS"*), face prompts (*e.g.*, *"Kanye West eating spaghetti"*), and amp prompts (*e.g.*, *"A Vietnam map, large motion"*) cannot be reliably detected through GPT-4o question-answering, therefore hard to proceed VIDEOREPAIR.

For evaluation metrics, we mainly adopt the average text-video alignment score they proposed. Among their all text-video alignment scores (CLIP-Score, SD-Score, BLIP-BLEU, Detection-Score, Count-Score, Color-Score, Celebrity ID Score, and OCR-Score) we exclude Celebrity ID Score and OCR-Score since they are related to 'face' and 'text' categories. Therefore, we calculate the text-video alignment score as $\text{Avg}(\text{CLIP-Score}, \text{SD-Score}, \text{BLIP-BLEU}, \text{DetectionScore}, \text{CountScore}, \text{ColorScore})$. For overall video quality, we directly adopt their metrics including Inception Score (Salimans et al., 2016) and Video Quality Assessment ($VQA_A$, $VQA_T$) (Wu et al., 2023a). For the motion quality score, we calculate the weighted average score of the Action Recognition score (from VideoMAE (Wang et al., 2023b)) and Average Flow score (Teed & Deng, 2020) from the official EvalCrafter code. For the temporal consistency score, we also calculate the weighted average score of Warping Error from optical flow (Wang et al., 2023b) and CLIP-Temp (Radford et al., 2021). For the *others* section of CogVideoX-5B, we report results on only 100 randomly sampled videos, as other baselines (e.g., OPT2I) require a significantly long refinement time (around 5h per one video refinement).

**T2V-Compbench.** Since VIDEOREPAIR has strength in compositional generation, we adopt T2V-Compbench (Sun et al., 2024a) and evaluate three dimensions: spatial relationships, generative numeracy, and consistent attribute binding. 'Spatial relationships' requires the model to generate at least two objects while maintaining accurate spatial relationships (*e.g.* 'to the left of', 'to the right of', 'above', 'below', 'in front of') throughout the dynamic video. 'Generative numeracy' specifies one or two object types, with quantities ranging from one to eight. 'Consistent attribute binding' contains color, shape, and texture attributes among two objects.

Following (Sun et al., 2024a), we adopt Video LLM-based metrics for consistent attribute binding and detection-based metrics for spatial Relationships and numeracy.

**Error Analysis.** We include screenshots of the evaluation questionnaire and labeling instructions in Figs. 13 and 14, respectively. For error analysis, we enlist three AI experts to assess the correctness of each step in the VIDEOREPAIR, including video evaluation, key object selection, and segmentation. For video evaluation, we present the initial text prompt and the corresponding video outputs generated by T2V-Turbo, displayed as a sequence of connected frames. In addition, we provide each question and its corresponding answer derived from the video evaluation results. Annotators are instructed to mark a generation as 'incorrect' if the number of generated objects does not exactly match the specified count in the initial prompt. For key object selection, we provide the object selection results produced by GPT-4o and ask annotators to verify whether the identified objects $O^*$ and $N^*$ are correct. For segmentation evaluation, we present a pointing prompt (e.g., "Point to the largest umbrella and one picnic blanket") along with the corresponding segmentation map, and ask annotators to judge whether the segmentation accurately aligns with the prompt.

## D ADDITIONAL QUANTITATIVE ANALYSIS

In this section, we present additional quantitative results to provide a deeper understanding. Specifically, we demonstrate that VIDEOREPAIR achieves superior efficiency in inference time compared to

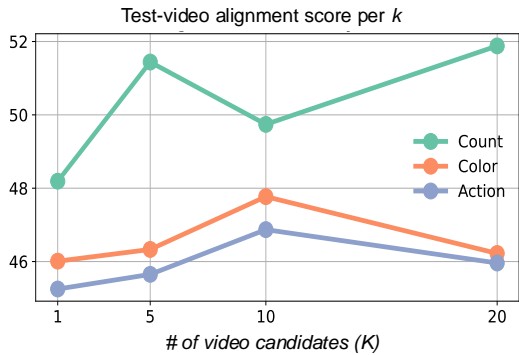

Figure 6: **Impact of the number of video candidates.** We vary the number of video candidates $K$ as 1, 5, 10, and 20 for ranking.

Table 6: **VIDEOREPAIR performance on the EvalCrafter *'Other'* section.** VIDEOREPAIR consistently improves video quality in camera movement, landscape, and style categories over the initial T2V-turbo generations.

|  | Camera movement | Landscape | Style |
|---|---|---|---|
| Initial video (T2V-turbo) | 44.02 | 48.94 | 42.70 |
| + VIDEOREPAIR | **45.23** | **50.71** | **43.63** |

other baselines, while also highlighting the impact of iterative refinement and the effect of varying the number of video candidates.

## D.1 INCREASING # OF VIDEO CANDIDATES

To evaluate the impact of video ranking, we vary the number of video candidates as $K = 1, 5, 10,$ and 20 during the ranking process. The variation among video candidates arises from different random seeds used to initialize $\epsilon'_0$. For example, video ranking is not applied when $K = 1$, and only one refinement is produced using a single random seed noise $\epsilon'_0$. For ranking metrics, we rely on the video score across all ablation studies. As depicted in Fig. 6, higher $K$ values (5, 10, and 20) consistently yield higher scores across all categories than $K = 1$. This trend is particularly prominent in the 'count' category, where increasing $K$ leads to noticeable performance improvements, highlighting the importance of considering multiple candidates for ranking.

## D.2 IMPACT OF GLOBAL REFINEMENT

Tab. 6 reports results on the *Other* section of EvalCrafter, which includes camera movement, landscape, and style prompts. Applying VIDEOREPAIR yields consistent improvements across all three categories, with gains of +1.21 in camera movement, +1.77 in landscape, and +0.93 in style. These results highlight that VIDEOREPAIR not only enhances core compositional attributes (e.g., count, color, action) but also extends effectively to broader aspects of video quality such as dynamics, scenery, and artistic style.

## D.3 IMPACT OF ITERATIVE REFINEMENT

We experiment with iteratively performing VIDEOREPAIR to further improve the text-video alignments. We monitor the video score and terminate the iterative refinement when it reaches 1.0 (max score), and use video ranking with K=5 candidates. As illustrated in Fig. 8, iterative refinement benefits all three prompt splits (count / color / action) of EvalCrafter. Additional iterative refinement examples are provided in Fig. 22.

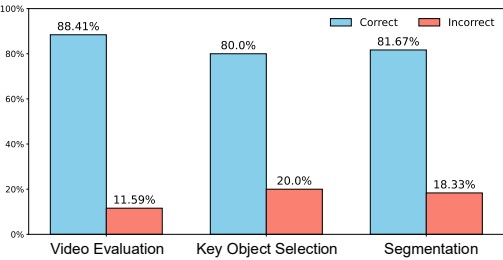

Figure 7: **Error analysis results.** VIDEOREPAIR consistently achieves approximately 80% correctness across all components of the framework.

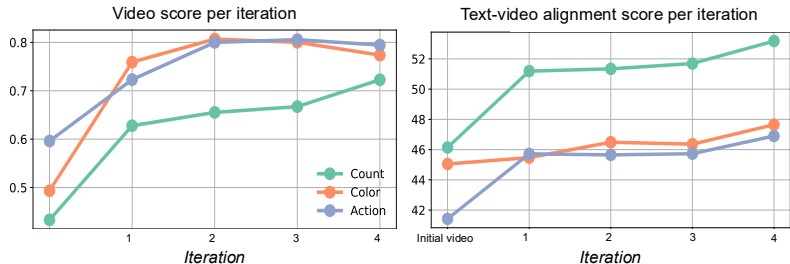

Figure 8: **Impact of iterative refinement.** Iterative refinement gradually improves the video score and text-video alignment score on all three prompt categories (count/color/action) of EvalCrafter. The 'initial video' refers to a video from T2V-turbo. We use video ranking with K=5 candidates.

### D.4    ERROR ANALYSIS OF VIDEOREPAIR.

We conduct human evaluations to assess the alignment of each step in VIDEOREPAIR with human judgments. Specifically, for video object evaluation, we present annotators with the initial video and corresponding MLLM-generated question-answer pairs, asking them to determine whether the answers are correct. For key object selection, we ask annotators to assess the correctness of the selected key object $O^*$. For pointing and segmentation, we evaluate whether the generated segmentation masks are well-localized with respect to the provided pointing prompts. As illustrated in Fig. 7, VIDEOREPAIR consistently achieves approximately 80% correctness across all components of the framework. While errors are observed at each step, we attribute these limitations to the current backbone model. We anticipate that future integration of more advanced backbones will lead to improved performance. The inter-annotator agreement for our evaluation is 93.18%, indicating strong consistency among human raters.

## E    ADDITIONAL QUALITATIVE EXAMPLES

**Moving key objects in VIDEOREPAIR.**    In long videos (e.g., CogVideoX-generated videos with 81 frames), key objects may disappear or newly appear across different frames. As shown in Fig. 10, VIDEOREPAIR effectively captures moving key objects $O^*$ using frame-wise masks $M$. This example illustrates how frame-wise masks help handle changes in object count and attributes - preserving disappearing objects (*car*) while incorporating previously missed objects (*house*).

### E.1    COMPARISON WITH BASELINES

We present additional qualitative comparisons with baseline methods (OPT2I (Mañas et al., 2024), SLD (Wu et al., 2024), and Vico (Yang & Wang, 2024)) in Figs. 15 to 21. These examples address a variety of failure cases commonly observed in T2V models, including inaccuracies in object count and attribute depiction, as highlighted in our main paper. Figs. 15 to 18 correspond to results from

T2V-Turbo, while Figs. 19 to 21 showcase examples from VideoCrafter2. Additionally, we provide binary segmentation masks that identify preserved areas (in black) and updated areas (in white).

Across these examples, VIDEOREPAIR effectively preserves the $O^*$ areas while refining the remaining regions using $p^r$. For instance, in Fig. 15, the camel from the original T2V-Turbo video is preserved, and a snowman is successfully added. In contrast, while SLD also leverages DDIM inversion to preserve objects, it often fails to integrate new objects seamlessly.

### E.2    ITERATIVE REFINEMENT

We also demonstrate the results of iterative refinement in Fig. 22, showing the initial video alongside the first and second refinements generated from T2V-Turbo. Overall, VIDEOREPAIR progressively enhances text-video alignment with each refinement step.

For numeracy-related cases (*e.g.*, six dancers and five cows), VIDEOREPAIR iteratively adds or removes specific objects, ensuring alignment with the given prompts. In cases of missing objects (*e.g.*, biologists and ducks), VIDEOREPAIR successfully generates additional biologists and multiple ducks while preserving the context of the initial video. Additionally, for attribute-related prompts (*e.g.*, yellow umbrella and blue cup), VIDEOREPAIR effectively refines object attributes, such as adding a wooden handle to the umbrella and enhancing the cup's blue color. These results demonstrate our ability to iteratively improve both object count and attribute alignment with high fidelity.

### E.3    OBJECT SELECTION IN VIDEOREPAIR

In step 2, we select the largest candidate among the correct objects. This approach can be seamlessly extended to select multiple correct objects when the number of objects in the initial video ($n_v^{o^*}$) meets or exceeds the prompt's specification ($n_p^{o^*}$). During the rebuttal, we implemented this extension to enable the formulation of **object-wise pointing prompts** and the generation of multiple masks to preserve these objects. As shown in Fig. 9, this version can preserve *a bear* and *a man* while automatically refining the video to add *an additional person*.

### E.4    STEP-BY-STEP ILLUSTRATION OF VIDEOREPAIR

In Figs. 11 and 12, we provide detailed illustrations of all three VIDEOREPAIR steps.

## F    LIMITATIONS

We introduce VIDEOREPAIR, a training-free and model-agnostic framework for evaluating and refining initially generated videos. Occasional hallucinations may arise in intermediate steps, as the overall performance is influenced by the quality of the underlying pre-trained backbones, including the LLM and MLLM. However, we note that VIDEOREPAIR is expected to improve further with future advancements in LLM and MLLM architectures.

## G    LICENSES

- VideoCrafter2
- EvalCrafter
- T2V-Compbench
- Semantic-SAM
- Molmo
- CogVideoX

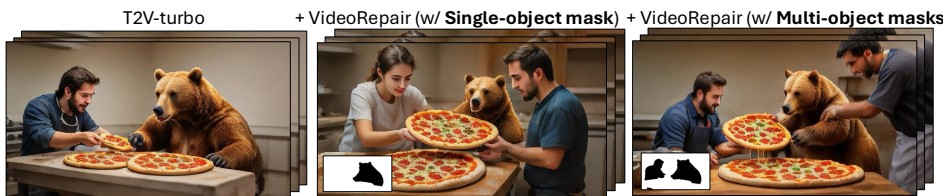

T2V-turbo  + VideoRepair (w/ **Single-object mask**)  + VideoRepair (w/ **Multi-object masks**)

*Prompt: 1 bear and 2 people making pizza*

Figure 9: **Single-object mask vs. Multi-object mask.**

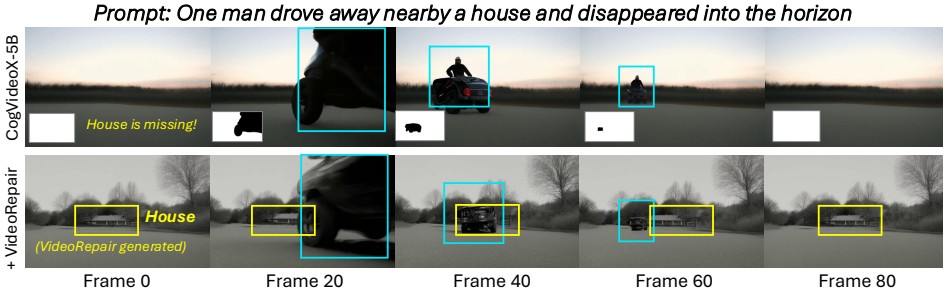

*Prompt: One man drove away nearby a house and disappeared into the horizon*

Figure 10: **Refining videos when the key object disappears.** VIDEOREPAIR successfully preserves disappearing objects (*car*) while incorporating previously missed objects (*house*).

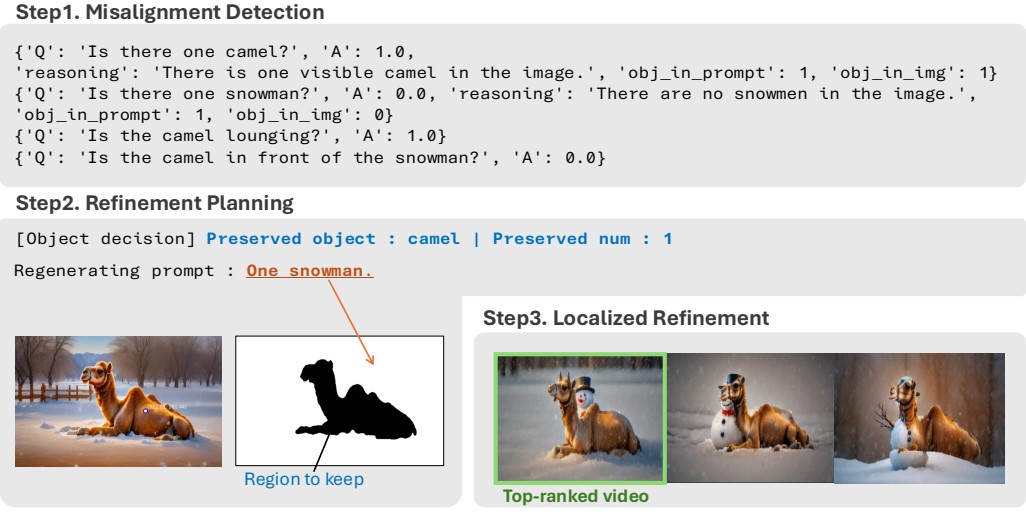

Figure 11: **Output from each step of VIDEOREPAIR.** We illustrate whole outputs from each step of VIDEOREPAIR.

**Step1. Misalignment Detection**

```
{'Q': 'Are there four children?', 'A': 1.0, 'reasoning': 'There are four visible children in the
image.', 'obj_in_prompt': 4, 'obj_in_img': 4}
{'Q': 'Are there three dogs?', 'A': 0.0, 'reasoning': 'There is only one dog visible in the image.',
'obj_in_prompt': 3, 'obj_in_img': 1}
{'Q': 'Is there a picnic?', 'A': 1.0}
{'Q': 'Is there a park?', 'A': 1.0}
{'Q': 'Are the children having a picnic?', 'A': 1.0}
{'Q': 'Are the children in the picnic?', 'A': 1.0}
{'Q': 'Are the dogs in the picnic?', 'A': 0.0}
{'Q': 'Is the picnic in the park?', 'A': 1.0}
```

**Step2. Refinement Planning**

[Object decision] **Preserved object : children | Preserved num : 4**

Regenerating prompt : Three dogs at a picnic in the park.

Region to keep

**Step3. Localized Refinement**

**Top-ranked video**

Figure 12: **Output from each step of VIDEOREPAIR.** We illustrate whole outputs from each step of VIDEOREPAIR.

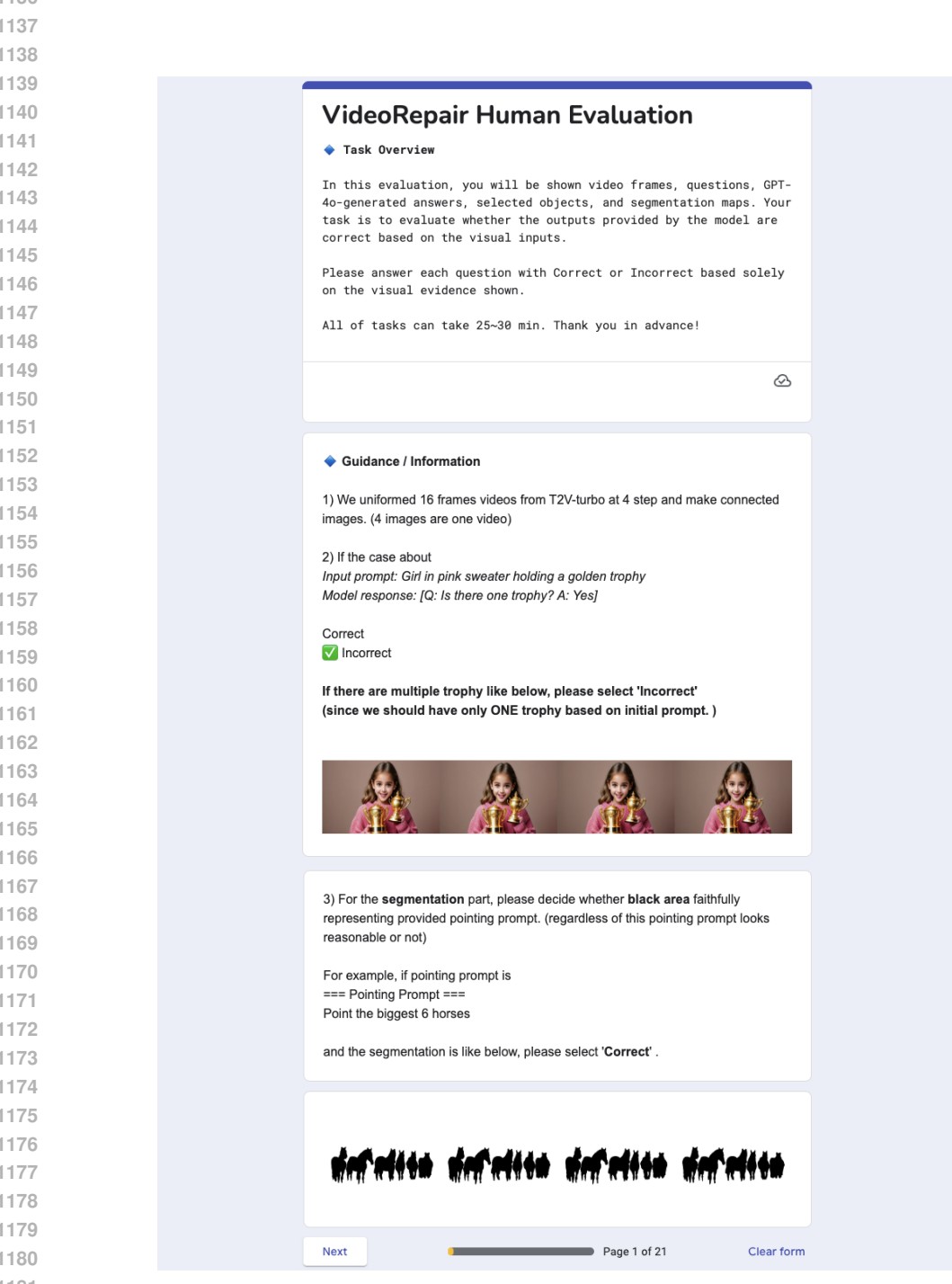

Figure 13: **A screenshot of questionnaires for error analysis.**

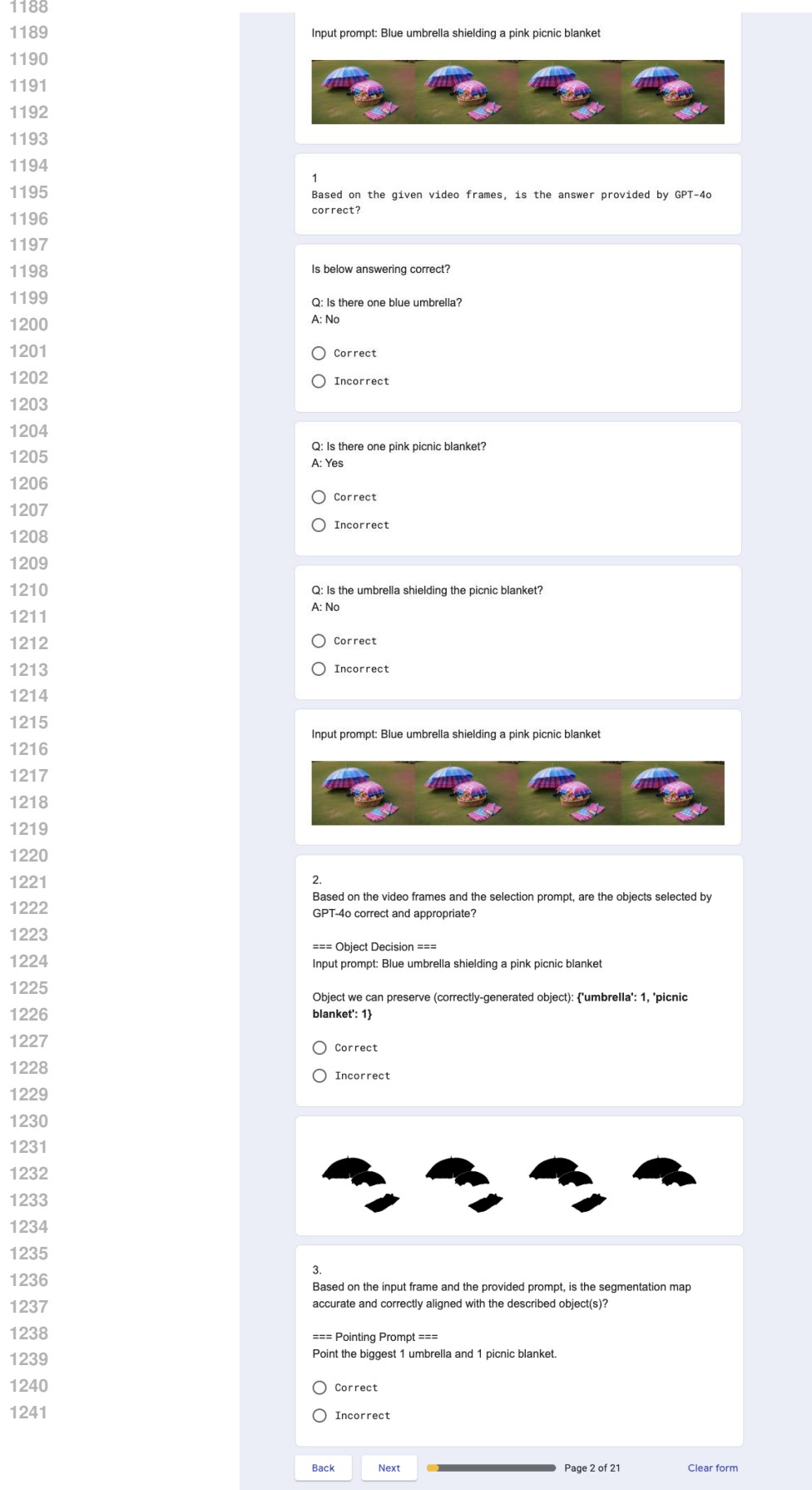

Figure 14: **A screenshot of questionnaires for error analysis.**

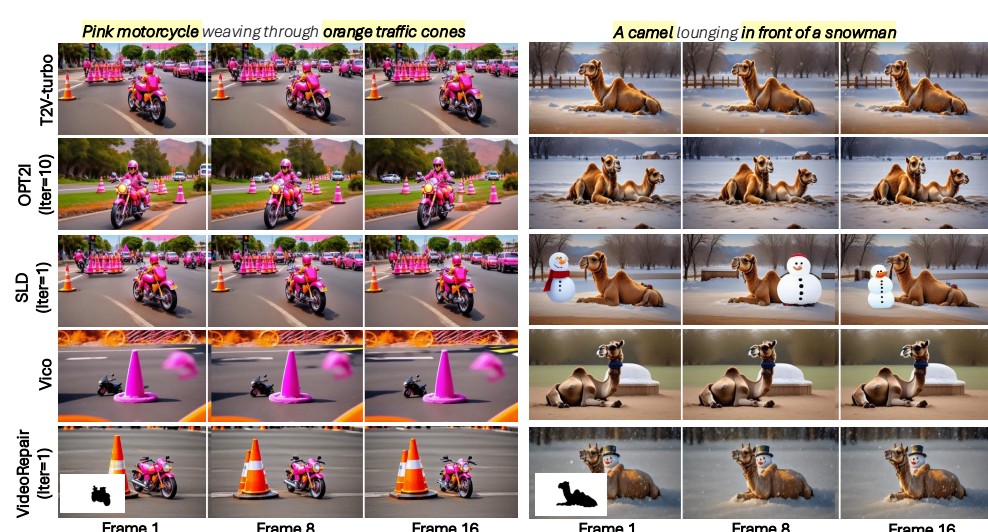

Figure 15: **Qualitative examples from T2V-turbo.**

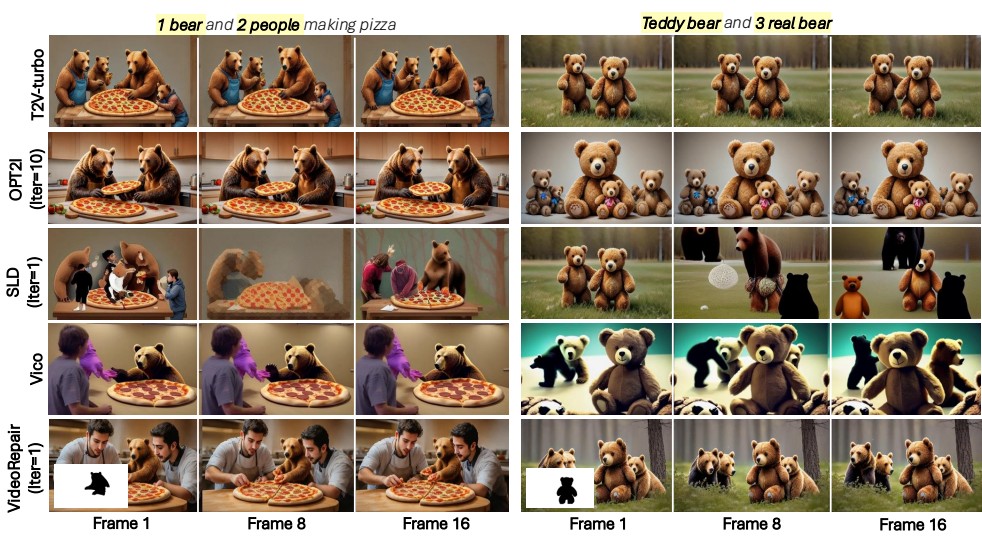

Figure 16: **Qualitative examples from T2V-turbo.**

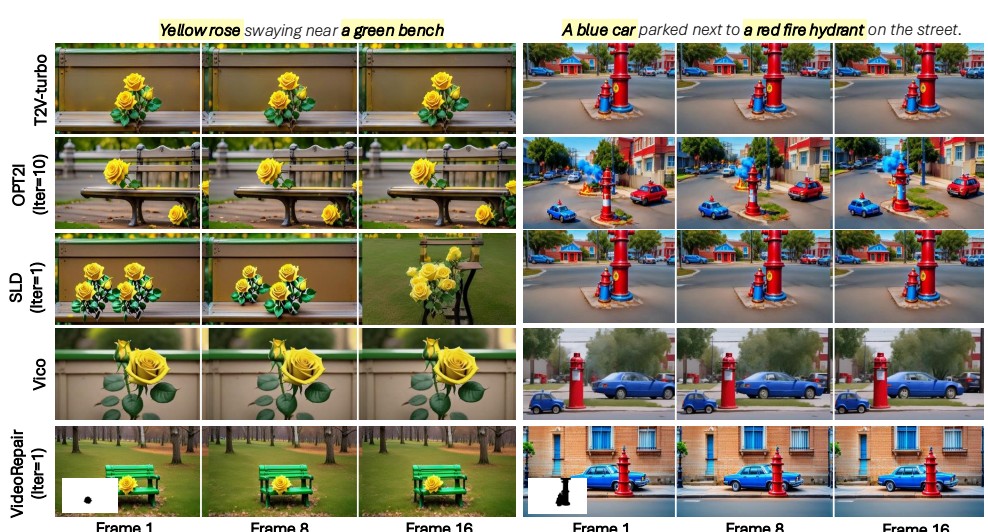

Figure 17: **Qualitative examples from T2V-turbo.**

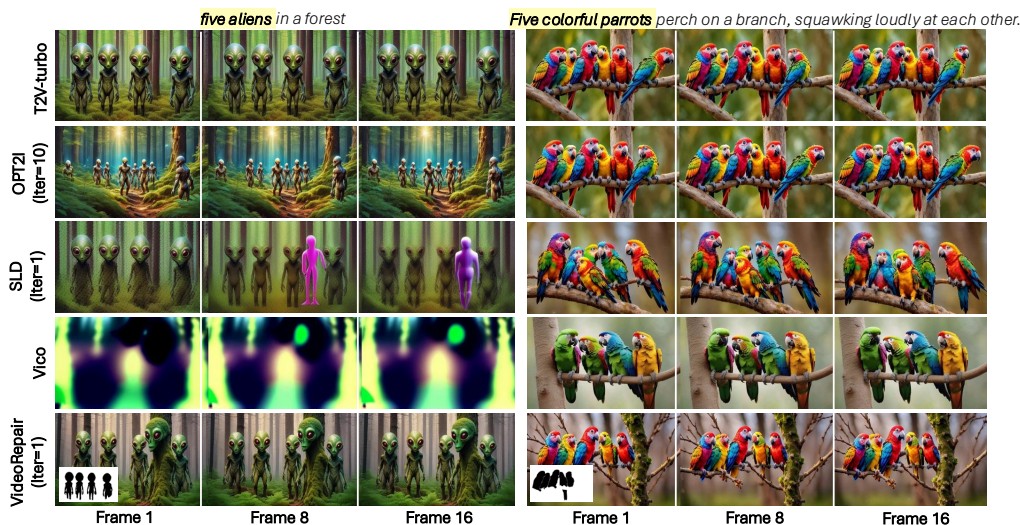

Figure 18: **Qualitative examples from T2V-turbo.**

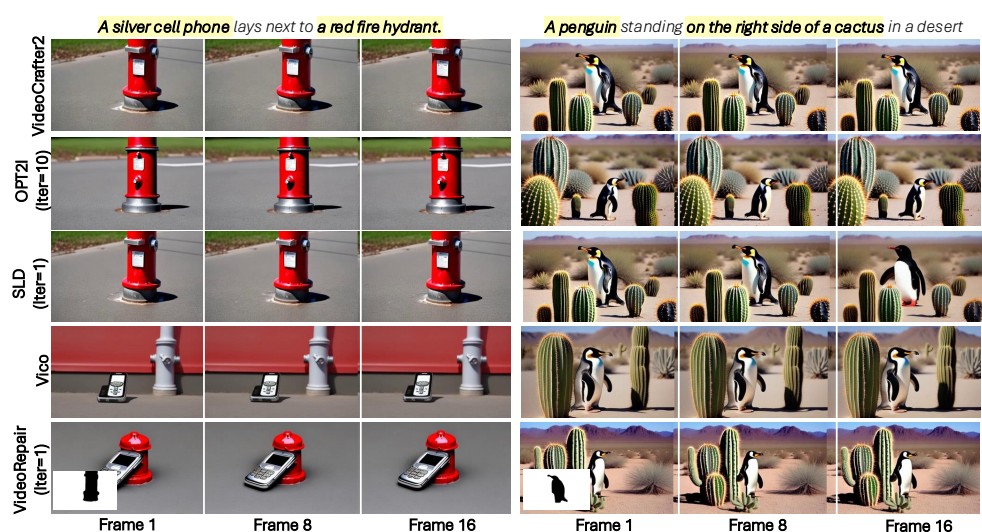

Figure 19: **Qualitative examples from VideoCrafter2.**

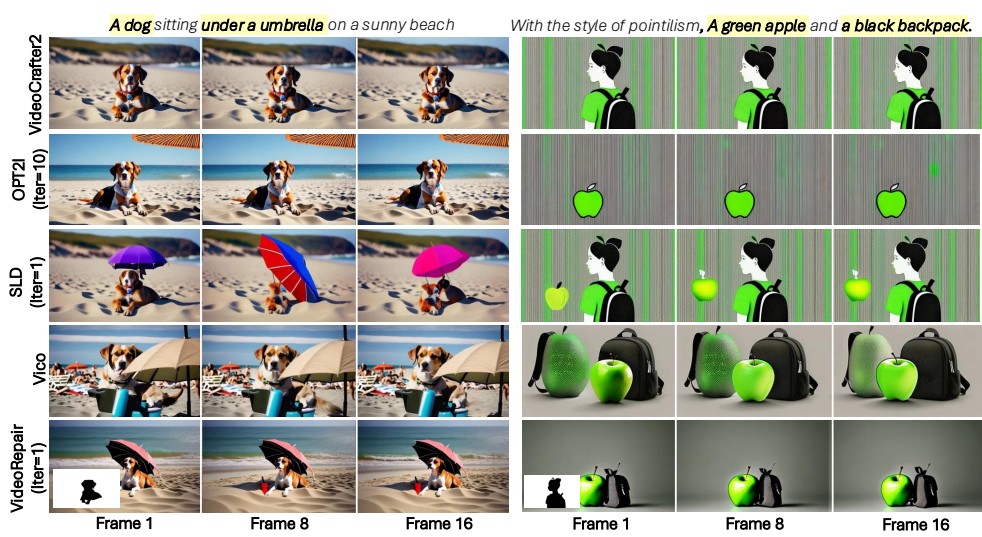

Figure 20: **Qualitative examples from VideoCrafter2.**

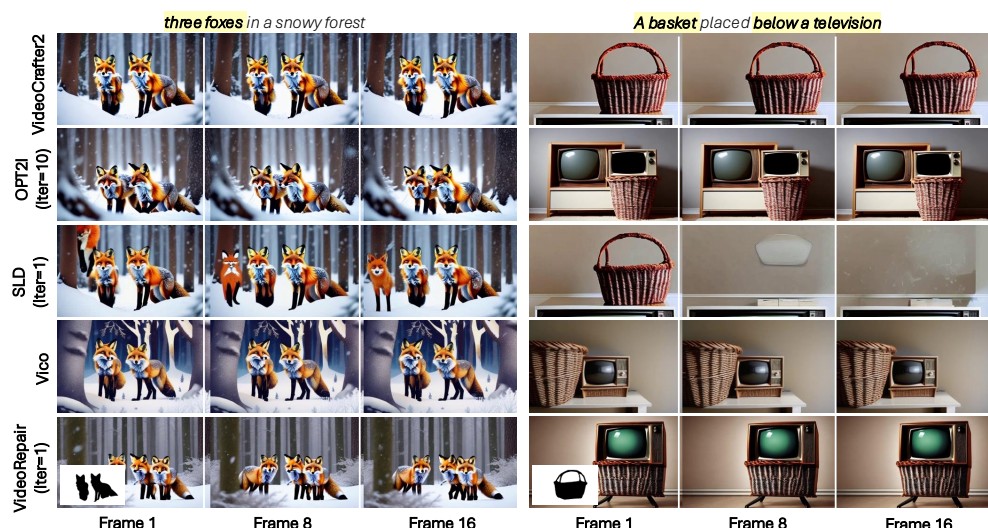

Figure 21: **Qualitative examples from VideoCrafter2.**

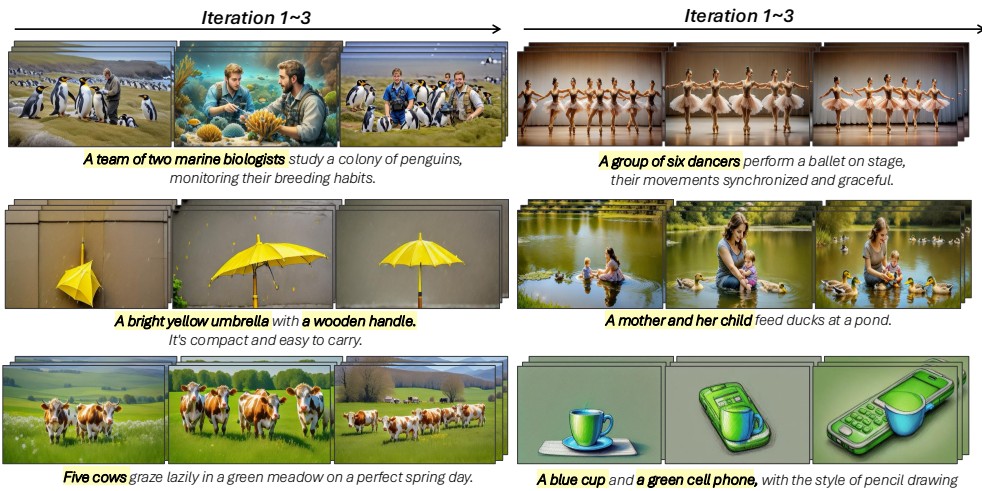

Figure 22: **Videos generated using iterative refinement with VIDEOREPAIR.** We depict iterative refinement results generated from T2V-Turbo. Overall, VIDEOREPAIR progressively enhances text-video alignment with each refinement step.

```
1. Given the question: "{cur_question}", provide a brief reasoning (up to two sentences) to determine the
accurate answer.
2. Respond to the question using binary values: 1.0 for "Yes" and 0.0 for "No". If the answer is uncertain
or unnatural due to image distortion or other issues, respond with 0.0 ("No").
3. Return the number of "{key_objects}" (as an integer) mentioned in the initial prompt "{cur_question}".
4. Return the number of "{key_objects}" (as an integer) in the provided image.

Return the result as a dictionary in the following format (not in JSON format):
{{"Q": "<question>",
"A": <binary answer>,
"reasoning": "<brief reasoning>",
"obj_in_prompt": <number of key object mentioned in the initial prompt>,
"obj_in_img": <number of key object in the image>}}

Example:
{{"Q": "Is there one robot?",
"A": 0.0,
"reasoning": "There are two visible robots in the image.",
"obj_in_prompt": 1,
"obj_in_img": 2}}

Please provide only the dictionary as the output without any additional text or explanation.
```

```
Respond to "{cur_question}" using binary values: 1.0 for Yes and 0.0 for No.
If the answer is uncertain due to image distortion or other issues, respond with 0.0 (No). \
Return the result as a dictionary in the following format (not in JSON format): \
{{"Q": "<question>", "A": <binary answer>}} \
(e.g., {{"Q": "Is there one robot?", "A": 0.0}}) \
Provide only the dictionary as the output, without any additional text or explanations.
```

Figure 23: **Prompts to perform visual question answering in video evaluation steps. Top:** The prompt for $Q_c^o$ (count-related question), **Bottom:** prompt for $Q_{\text{others}}^o$ (attribute-related question). `cur_question` means each video evaluation question and `key_objects` means entity word in each question.

```
Given the image which compose of multiple concatenated frames from a video and the list of question-
answer pairs for each object, represented as {object_wise_dict}, choose all the accurately or visibly
generated objects from the list {objects_from_Question}. Prioritize selecting objects with a high
number of answers rated 1.0 for each question. Select the object that is both large and clearly
visible, prioritizing prominent objects (such as animals, humans, or specific items) over background
elements (like ocean or city). Return only the name of the best object to keep from the list, without
additional explanation (e.g., dog)
```

Figure 24: **Prompt to choose which object(s) to preserve.** We ask GPT4o to select objects to preserve in the scene.

```
Given the following list of questions {question_list}, create a single descriptive sentence that combines
the meaning of each question into a natural, affirmative statement that provides a full, concise summary.

Examples:
- Example 1
Question list: ['Is there a bed?', 'Is the bed blue?', 'Are the pillows beige?', 'Are the pillows with the
bed?']
Answer: "Blue bed with beige pillows."

- Example 2
Question list: [Are there three real bears?]
Answer: "Three real bears."

- Example 3
Question list: [Are there two people?, Are the people making pizza?]
Answer: "Two people making pizza.

- Example 4
Question list: [Is there a family?, Is there one cat?, Is there a park?, Is the family taking a walk?, Is
the cat walking?, Is the family enjoying?, Is the family breathing fresh air?, Is the family exercising?]
Answer: "A family and a cat are walking in the park."

- Example 5
Question list: [Is there a green bench?, Is there an orange tree?, Is the bench green?, Is the tree
orange?]
Answer: "Green bench and orange tree."

Your Current Task: Your response should be a concise 1 phrase, without additional explanation
(e.g., "a small bear")
```

Figure 25: **Prompt to plan how to refine the other regions.** We use five in-context examples to create the refinement prompt from the question related to other objects.

```
Generate 1 paraphrase of the following image description
while keeping the semantic meaning: "{init_prompt}".
Provide your response as a single phrase without any explanation.
Format it as: <PROMPT> ... </PROMPT>.
(e.g., <PROMPT>Two dogs and a whale embark on a sea adventure.</PROMPT>)
```

Figure 26: **Prompt for LLM paraphrasing.** Following OPT2I (Mañas et al., 2024), we ask GPT4 to generate diverse paraphrases of each prompt for LLM paraphrasing baseline experiments.

