# OpenReview forum: "Self-Correcting Text-to-Video Generation with Misalignment Detection and Localized Refinement"
_ICLR.cc/2026/Conference — ICLR 2026 Conference Withdrawn Submission_

### Official Review · Reviewer_EeWe · 2025-10-30

**Soundness:** 3
**Presentation:** 3
**Contribution:** 2
**Rating:** 2
**Confidence:** 4

**Summary:**

This paper introduces VIDEOREPAIR, a video refinement framework to correct text-video misalignments. It has three steps: 1. detect misalignment. Finding the issue and region with MLLM. 2. Plan the refinement including preserve the correct parts and construct prompts that could be used to re-generate the target parts. 3. regenerate the incorrect parts.
The method is evaluated on two benchmark EvalCrafter and T2V-CompBench on three different text to video models.

**Strengths:**

1. The method is intuitive: finding the incorrect part and fix it.
2. The author demonstrate the performance with different models on different benchmarks

**Weaknesses:**

1. The video quality in the paper and supplementary material is not of high quality. Although it fixed the alignment issue, the quality is far below sota video generation model. Meanwhile, the method is not tested on sota models.

2. The blending process could be hard for object parts edit. For example, if a shirt or eye color is incorrect, then it is hard for the aggregating pipeline. While the author did not show those results.

3. While the method claim to be model agnostic, but many component like noise initalization is based on diffusion-based model. In other words, it still only fit certain architecture.

**Questions:**

1. what are the cost of the total editing process?

2. For the same cost or the same iteration, if we regenerate the video the same time or the same cost, will that yeild a better performance?

---

### Official Review · Reviewer_hMBu · 2025-10-31

**Soundness:** 2
**Presentation:** 2
**Contribution:** 1
**Rating:** 2
**Confidence:** 4

**Summary:**

To address the challenge that current text-to-video (T2V) models often fail to align with complex text prompts，the authors propose VideoRepair, a training-free, self-correcting, and model-agnostic video refinement framework. VideoRepair automatically detects fine-grained text–video misalignments and performs targeted, localized corrections. The key contributions are as follows:
- Misalignment detection, which identifies both faithful and misaligned regions within generated videos;
- Refinement planning, which preserves correctly generated entities, segments their corresponding regions across frames, and constructs targeted prompts for misaligned areas;
- Localized refinement, which selectively regenerates problematic regions while preserving faithful content through joint optimization of preserved and newly generated areas.

**Strengths:**

- The motivation of the work is clear and well-justified.
- The experimental evaluation is extensive.

**Weaknesses:**

- The authors point out that existing methods suffer from high computational cost, visual inconsistency, and temporal incoherence. However, they do not provide sufficient analysis or discussion on how their proposed approach addresses these issues.
- It remains unclear how the proposed method ensures adaptability and interpretability across different prompts.
- As is well known, large language models (LLMs) are prone to hallucinations. It is unclear how the authors ensure the correctness of the generated results. For example, when the method relies on prompts to obtain 2D coordinates, how do the authors verify that the predicted locations are accurate?
- It is not clearly stated what type of segmentation model is used to extract the masks. I think the mask is much important for the results in such mask-guided refinement strategy.
- From the refined video shown in Figure 1, the newly generated person still exhibits noticeable distortions, particularly in the hands, which appear unrealistic. Moreover, the visualizations in Figure 3 seem unfaithful, different frames show significant inconsistencies in object appearance (e.g., the pigs), suggesting a lack of temporal smoothness. The authors also did not provide any video demonstrations to substantiate the claimed temporal consistency. Based on my experimental experience, such a refinement strategy may introduce undesirable side effects, such as noticeable jitter or flickering in the generated videos.
- The authors claim to provide spatio-temporal feedback signals; however, they do not clearly explain how the spatio-temporally consistent queries are generated or formulated within their framework.
- In Table 3, it is difficult to assess the effectiveness of the proposed module. For example, the first row should include a comparison between results with and without the evaluation component to demonstrate its contribution. However, the authors do not provide such ablation results, nor do they include corresponding visual examples for validation. As a result, it is hard to convincingly support the effectiveness of the proposed solution.

**Questions:**

see weaknesses

**Details Of Ethics Concerns:**

The core issue is that the reported metrics are insufficient to demonstrate the effectiveness of the proposed method, as the paper lacks extensive visual and video evidence to support its claims.

---

### Official Review · Reviewer_sEDb · 2025-10-31

**Soundness:** 2
**Presentation:** 3
**Contribution:** 2
**Rating:** 4
**Confidence:** 4

**Summary:**

This paper addresses the text-video misalignment problem under complex cues in T2V generation by proposing a model-agnostic, training-free, refined framework, VIDEOREPAIR. Its core achieves self-correction through a two-stage process: first, it utilizes a multimodal large model (MLLM) to generate fine-grained spatiotemporal problem detection, identifying misaligned regions and locking in the correct content; then, through region-preserving segmentation and target cue construction, it locally regenerates the problem region and integrates the global content.

**Strengths:**

1. This paper introduces a novel region-preserving self-correction paradigm for T2V refinement. The integration of MLLM-driven spatio-temporal error detection with localized noise re-initialization and prompt decomposition is a creative combination of existing ideas tailored to the unique challenges of video (temporal coherence, frame-wise consistency).
2. VideoRepair’s training-free and model-agnostic nature makes it easily deployable with existing diffusion-based T2V backbones.

**Weaknesses:**

1. The core misalignment detection and planning steps rely heavily on MLLM's performance.
2. Iterative refinement design is underdeveloped: The authors briefly mention iterative refinement but provide limited details on its practical utility. For example, there is no analysis of how many iterations are typically needed for different prompt types, whether performance plateaus after a certain number of steps, or how to balance iteration count with inference efficiency.
3. The motivation is somewhat trivial, only considering the issue of subject identity and using masks and segments to handle it. However, video generation has many other problems, such as poor animation and image degradation, which were not taken into account.

**Questions:**

1. How does VIDEOREPAIR handle complex failure modes such as (a) object occlusion across frames, (b) motion blur leading to ambiguous object boundaries, and (c) conflicting prompt constraints? Could you provide qualitative examples and quantitative metrics for these scenarios, and explain how the current framework addresses (or fails to address) them?
2. What is the typical number of iterations required for iterative refinement across different prompt categories (count, spatial relations, attribute binding)? Can you show a performance-efficiency curve (alignment score vs. number of iterations vs. inference time) to guide practical use?
3. Could you clarify how the framework handles prompts with implicit spatial/temporal relations (e.g., "a cat chasing a mouse from left to right over 10 frames")

---

### Note · Authors · 2025-11-12

I have read and agree with the venue's withdrawal policy on behalf of myself and my co-authors.